

# Novel trajectory clustering method based on distance dependent Chinese restaurant process

Reza Arfa[1,2], Rubiyah Yusof[1,2] and Parvaneh Shabanzadeh[1,2]

[1] Centre for Artificial Intelligence and Robotics, Universiti Teknologi Malaysia, Kuala Lumpur, Malaysia
[2] Centre for Artificial Intelligence and Robotics, Malaysia-Japan International Institute of Technology (MJIIT), Universiti Teknologi Malaysia, Kuala Lumpur, Malaysia

## ABSTRACT

Trajectory clustering and path modelling are two core tasks in intelligent transport systems with a wide range of applications, from modeling drivers' behavior to traffic monitoring of road intersections. Traditional trajectory analysis considers them as separate tasks, where the system first clusters the trajectories into a known number of clusters and then the path taken in each cluster is modelled. However, such a hierarchy does not allow the knowledge of the path model to be used to improve the performance of trajectory clustering. Based on the distance dependent Chinese restaurant process (DDCRP), a trajectory analysis system that simultaneously performs trajectory clustering and path modelling was proposed. Unlike most traditional approaches where the number of clusters should be known, the proposed method decides the number of clusters automatically. The proposed algorithm was tested on two publicly available trajectory datasets, and the experimental results recorded better performance and considerable improvement in both datasets for the task of trajectory clustering compared to traditional approaches. The study proved that the proposed method is an appropriate candidate to be used for trajectory clustering and path modelling.

## INTRODUCTION

The trajectory of a moving object obtained by tracking the object's position from one frame to the next is a simple yet efficient descriptor of an object's motion. Trajectory analysis has long been a research focus in different fields of study (*Jonsen, Myers & Flemming, 2003*; *Pao et al., 2012*; *Reed et al., 1999*; *Fox, Sudderth & Willsky, 2007*). In the context of intelligent surveillance systems (ITS) (*Tian et al., 2017*), trajectory clustering is a critical core technology in many surveillance applications including activity analysis (*Morris & Trivedi, 2011*), path modelling (*Zhang, Lu & Li, 2009*), anomaly detection (*Dee & Velastin, 2008*), and road intersection traffic monitoring (*Aköz & Karsligil, 2014*).

Many trajectory analysis systems consist of two main steps. In the first step, trajectories are grouped into clusters based on their similarities. Most proposed methods assume the number of clusters to be known. After the trajectories are clustered, the path taken by agents

Corresponding authors
Reza Arfa, rezaarfa@gmail.com
Rubiyah Yusof, rubiyah.kl@utm.my

in each cluster will be modelled. There are at least two limitations with these approaches. First, in real-world problems, the number of clusters is usually unknown or is expensive to acquire. Furthermore, trajectory clusters and path models are closely related, whereby the knowledge of one helps in improving the performance of the other.

Most existing trajectory analysis methods can be categorized into similarity-based models and Probabilistic Topic Models (PTM). The main stages of similarity-based approaches are calculating a similarity matrix and clustering the trajectories based on the similarity matrix. At the first stage, pairwise similarities between trajectories are obtained via a similarity function and stored into a $N \times N$ matrix, where $N$ is the total number of available trajectories. Defining a suitable similarity measure is a challenging task that directly affects the overall accuracy of the system (*Zhang, Kaiqi & Tieniu, 2006*). Well-known similarity measures used for trajectory analysis include Euclidean distance, dynamic time wrapping (DTW) (*Keogh & Pazzani, 2000*), Hausdorff distance (*Atev, Miller & Papanikolopoulos, 2010*), and Longest Common Sub-Sequences (LCSS) (*Vlachos, Kollios & Gunopulos, 2002*). After the similarity matrix is obtained, the second stage uses any standard clustering algorithm to cluster the trajectories into $K$ clusters based on their similarities. Typical clustering algorithms include spectral clustering (*Ng, Jordan & Weiss, 2002*), agglomerative clustering (*Xi, Weiming & Wei, 2006*), and fuzzy c-means (*Weiming et al., 2006*). The main disadvantage of similarity-based approaches is that it requires the number of clusters, $K$, to be known in advance.

When trajectories are clustered, some studies perform path modelling in a further stage. Path models are useful in intelligent surveillance systems and used for compact representation of clusters, performing real-time anomaly detection (*Morris & Trivedi, 2011*), and high-level scene understanding (*Lei et al., 2014*), and route planning (*Joseph et al., 2011*). *Makris & Ellis (2005)* modelled the path as an envelope, which denotes the extent of a path by finding the two farthest samples in a cluster. *Morris & Trivedi (2011)* used the weighted average of trajectories of each cluster to form the path model for that cluster. Based on the dominant set clustering approach, *Yiwen et al. (2014)* proposed a system that obtains the scene structure from clustered trajectories.

All these approaches, however, model the path after the trajectories are clustered. Therefore, the performance of the modelled path is limited to how well trajectories are clustered. Also, the modelled path is not used to improve the trajectory clustering.

Another well-known class of approaches in trajectory analysis is based on probabilistic topic model (PTM) (*Papadopoulos, 2008*). In PTM approaches, trajectories are first converted into a set of symbols via a pre-defined codebook. This new representation of trajectories is then treated as documents while the symbols are treated as words. Compared to a similarity-based approach, trajectory analysis methods based on PTM do not usually require the number of clusters in advance.

*Jeong, Chang & Choi (2011)* used latent Dirichlet allocation (LDA) and the hidden Markov model (HMM) to discover the semantic regions and the temporal relationship between them. A two-level LDA topic model is proposed by Song et al. (*Lei et al., 2014*). The first level LDA models the motion of single-agent as distributions over patch-based

features. The second level LDA uses the output of the first-level to learn interactions over multi-agents. This model, however, does not perform trajectory clustering.

*Wang et al. (2011)* proposed a dual hierarchical Dirichlet process (Dual-HDP). Unlike previous PTM models, Dual-HDP is capable of clustering the trajectories and modelling the semantic scene at the same time. Each semantic region is modelled as a distribution over grids, and the scene is modelled as a distribution over the semantic regions. The number of clusters and the semantic scene is decided automatically. Since the model relies only on bag-of-grids representation, it cannot capture the long-term dependency between observations. This results in having a partial path model for each cluster. Having a full path model is an important step for interpreting agents' movement in scenarios such as highways and junctions.

Furthermore, since only quantised trajectories are used, the overall performance of Dual-HDP is highly sensitive to grid size. Choosing a large grid size rapidly decreases the performance due to quantisation error. On the other hand, choosing a small grid size requires considerably more amount of data to learn the trajectory patterns.

This study proposed a trajectory clustering and path modelling system that clusters the trajectories and models the path taken by each cluster at the same time. Our approach is based on distant dependent Chinese restaurant process (DDCRP) (*Blei & Frazier, 2011*), which is a generalisation of the normal Chinese restaurant process (CRP) (*Pitman, 2002*).

## METHODS

### Distance dependence chinese restaurant process

The Chinese restaurant process (CRP) is a distribution on partitions of integers proposed by *Pitman (2002)*. CRP can be explained by the following analogy: Imagine a Chinese restaurant with an infinite number of tables. The first customer enters the restaurant and sits at the first table with probability1. Next, customers enter the restaurant and sit at occupied tables with probability proportional to the number of customers sitting on that table or sit at an empty table with the probability relative to a parameter $\alpha$. After this process, which is known as a customer-table assignment, customers sitting on the same table will share a similar dish. This process can be described as follows:

$$P(z_i = k | z_{-i}, \alpha) \propto \begin{cases} n_k, k \leq K \\ \alpha, k = K+1 \end{cases} \tag{1}$$

where $z_i$ denotes table assignment for the $i$th customer, $K$ is the total number of occupied tables, and $z_{-i}$ is table assignmthe ent of all other customers except $i$th customer, and $n_k$ is the total number of customers sitting on the $i$th table. More details of CRP and its connection to Dirichlet process can be found in *Gershman & Blei (2012)*.

The distance dependence Chinese restaurant process (DDCRP) generalises the CRP and allows for a non-exchangeable distribution over partitions (*Blei & Frazier, 2011*). Unlike CRP, where each customer is assigned to a table, in DDCRP each customer is assigned to another customer with a probability relative to their distance/similarity. Therefore, the more similar two customers, the more probable they will get a direct link. It is important to

note that it is still possible for two customers with small similarities to be indirectly linked to each other via intermediate customers. After this procedure, which is also known as a customer to customer assignment, customers who are directly or indirectly linked will sit down at a table and share a similar dish.

More formally, let $d_{ij}$ represent the distance between $i$th and $j$th customers. Probability of customer $i$ have a direct link with customer $j$ is calculated as:

$$P\left(c_i = j | \boldsymbol{D}, f, \tau\right) \propto \begin{cases} \tau, & if \quad i = j \\ f(d_{ij}), & otherwise \end{cases} \tag{2}$$

where $f(d)$ denotes a monolithically decreasing decaying function that satisfies $f(\infty) = 0$, $\boldsymbol{D}$ is the matrix of pairwise distance between customers, and $\tau$ is a constant that indicates the probability of self-link.

The DDCRP was proposed originally for modelling non-exchangeable text documents where the distance between the dates of documents determines their similarity. The documents are converted into their bag-of-words (BoW) representation before the posterior probability of DDCRP is calculated. Such a conversion to BoW representation is a crucial step that makes the inference of DDCRP computationally tractable.

Recently researchers have adopted DDCRP for problems beyond language processing. *Ghosh et al. (2011)* proposed a hierarchical extension of DDCRP for producing coarser image segmentations in the form of human-like segmentations. In a more recent study, *Baldassano, Beck & Li (2015)* used DDCRP to model a complex web of connections with a small number of interacting units. The proposed method is used to model the connectivity between sub-regions of the human brain and analysing human migration behaviour. Also, *Ren et al. (2016)* used DDCRP for key frame selection from unordered image sets, where the selected frames are used for dense 3D reconstruction.

## Trajectory analysis with distance dependent CRP

Unlike text data where observations in documents are words sampled from a corpus with a limited number of words, observations in trajectories are not discrete. Trajectories are vectors with varying length where each observation gets a real value bounded by the scene's size. One can divide the scene into blocks of equal sizes and convert a trajectory into its discrete form. After such a conversion, the resulting quantized trajectories are equal length vectors and each observation gets a discrete value. The size of grids in this case, however, will have a direct impact on the system performance. While theoretically smaller grids can improve the performance, they require substantially more data for training.

Another disadvantage of treating trajectories as documents is the bag-of-words representation. Such representation discards the order between observations. Discarding the orders between samples in trajectory data is problematic since it is possible for agents from opposite directions to share the same observations over grids. One solution to avoid this problem is to quantise the direction of observations (*Wang et al., 2011*). Estimating the direction of observation requires further processing and sometimes includes an inaccurate estimation. Such a quantisation increases the size of the corpus and, therefore, requires more data for training. In addition, with bag-of-word representation alone long-term

dependencies between observation cannot be captured which results in having partial path models in existing PTM approaches.

We addressed these problems by using similarity between trajectories as the prior probability in DDCRP. Using such a prior probability limits the assignment of trajectories and promotes trajectories to get linked based on how similar two trajectories are. In addition to the similarity measure, whether the trajectories are linked together or not, also will depend on their discrete observation over the grids. Since most similarity measures can be applied prior to converting the trajectory into discrete form, such a formulation is less sensitive to the choice of grid size. In addition, since some similarity measures, including Modified Hausdorff and LCSS, also take the order of the observations into account, it is not required to quantise the direction anymore.

Any raw trajectory $T_i$, is usually represented by a sequence of its $n_i$ observation $T_i = [o_{i,1}, ..., o_{i,l}, ..., o_{i,n_i}]$. In this representation, $o_{i,l}$ indicates $l$th observed position of $i$th object. Let $d_{ij}$ to indicate pairwise distance between $i$th and $j$th trajectories. This distance can be of any general distance used to measure similarity between trajectories. The result of pairwise distance between $N$ trajectories can be stored in a distance matrix and denoted as $D \in \Re^{N \times N}$.

Apart from the calculation of distance matrix discussed above, raw trajectories are converted into bag-of-grids representation. For this, the scene is divided into M grid cells of equal size. Based on the cell in which it falls into each observation of a trajectory $o_{i,l}$, is individually quantised. Then a raw trajectory, $T_i$, is approximated by bag-of-grid represetnation $X_i \in \Re^M$. Each element of $X_i(s)$ indicates the number of times $i$th trajectory had an observation in the $s$th grid cell.

Using DDCRP's metaphor, we use the bag-of-grid representation of trajectories as customers, clusters as the tables and path models as dishes. Based on the definition of DDCRP, it is not possible to draw the table directly. Instead, the outgoing link for each customer needs to be drawn. Trajectories that directly or indirectly link together are considered to be in the same cluster. All trajectories in the similar cluster share the same path model which is a multinomial distribution over the grid cells. Each path model is independently drawn from a base distribution $G_0$. In our case, $G_0$ is a Dirichlet distribution. The full generative process for the news program is as follows:

1. For each trajectory, sample customer assignment $C_i \sim ddCRP(D, f, \tau)$ as explained in Eq. (2).
2. Drive table assignment from customer assignment. For each table, $k$, sample its parameter from the base distribution $\varphi_k \sim G_0$
3. For each trajectory, independently draw $X_i \sim Mul(.|\varphi_{z_i})$
   The decaying function, $f(.)$, in Eq. (2) was defined as:

$$f(d; \gamma; \gamma_0) = \exp(-\frac{d}{\gamma}).$$ (3)

With this function, the probability of linking two trajectories becomes smaller as their distance increases. The parameter $\gamma$ controls how fast this probability decays with increasing distance. The inference of DDCRP requires drawing samples for all samples which have the possibility of being linked.

## Inference

The key problem that needs to be addressed is computing the posterior distribution of latent customer assignment conditioned on the bag-of-grid cell representation of trajectories $X_{1:N}$. In our problem, the based distribution $G_0$, is conjugate to the data generating distribution $P(X_i|Z_{c_i}, G_0)$. Therefore, the cluster parameters $\varphi_k$ can be analytically marginalised. After such a calculation, the posterior distribution is expressed by:

$$P\left(c_{1:N}|X_{1:N},D,f,\tau,\gamma,\gamma_0\right) \propto \prod_{i=1}^{N} P(c_i|D,f,\tau,\gamma,\gamma_0)P(X_{1:N}|Z(c_{1:N})) \tag{4}$$

where $Z(c_{1:N})$ denotes the table assignment and $P(X_{1:N}|Z(c_{1:N}))$ is the likelihood function which can be expressed by *Blei & Frazier (2011)*

$$P(X_{1:N}|Z(c_{1:N})) = \prod_{k=1}^{|Z(c_{1:N})|} P(X_{z^k(C_{1:N})}|Z(c_{1:N})) \tag{5}$$

with $|Z(c_{1:N})|$ being the number of unique tables and $z^k(C_{1:N})$ denoting all customers assigned to table $k$.

Due to the combinatorial sum in the denominator, the analytical solution of the posterior given by Eq. (4) is intractable. Instead of exact inference, collapsed Gibbs strategy (*Blei & Frazier, 2011*) is used to derive the posterior inference where the customer assignment is iteratively sampled from the following equation:

$$P(c_i|c_{-i},X_{1:N},D,f,\tau,\gamma,\gamma_0) \propto P(c_i|D,f,\tau) \times P(X_{1:N}|z(c_i \cup c_{-i})) \tag{6}$$

where $c_{-i}$ denotes all customer assignments except for $c_i$. The first term on the right side of the equation is DDCRP's customer assignment discussed in Eq. (2), and the second term is the likelihood term given by Eq. (5). More details can be found in the Supplemental Material.

## RESULTS AND DISCUSSION

The performance of the proposed approach was evaluated on the CROSS (*Morris & Trivedi, 2011*) and the Lankershim datasets (*NGSIM: Next Generation Simulation, 2008*).

The CROSS dataset provides objects trajectories and their ground truth activities. The data are organized into train and test sets. There are 1,900 and 9,700 trajectories in the train and test sets respectively. Two hundred samples in the test set are labeled as abnormal activities. These samples were discarded in this study and we evaluated the proposed model on 9,500 trajectories in the test set with legal activities (Fig. 1).

The Lankershim dataset is part of the Next Generation Simulation (NGSIM) program provided by the US Federal Highway Administration (FHWA). The dataset contains videos taken with overhead intersection cameras. The dataset also provided the trajectories of moving vehicles. Based on the time the videos are collected, the data are placed into 8:30 am to 8:45 am and 8:45 am to 9:00 am subsets. The trajectories took place near an intersection, and trajectories outside of this area were removed (see Fig. 2). The corresponding X and Y coordinate for this region were $-80 < X < 80$ and $300 < Y < 500$ respectively. After

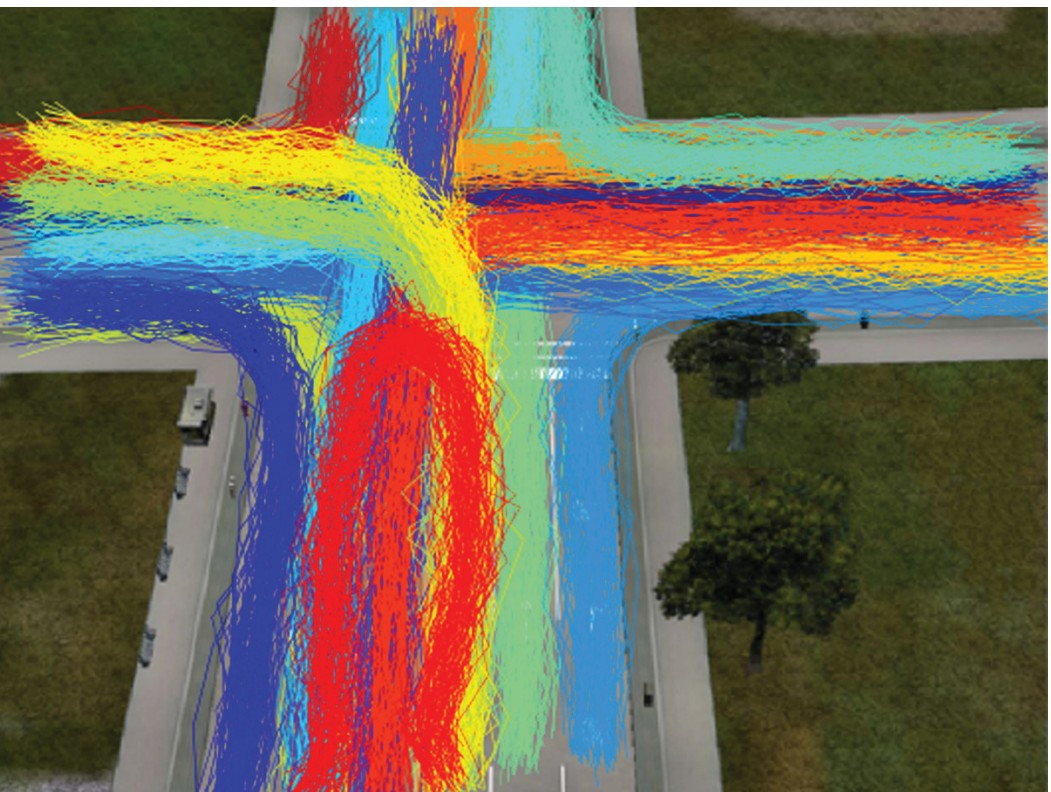

**Figure 1  Vehicle trajectories in CROSS dataset.** The colors of trajectories indicate the ground truth activity label.

filtering the trajectories having less than ten observations, a total of 2212 trajectories were obtained. Since this dataset does not provide activity labels for trajectories, the trajectories were manually labelled into 21 activities (19 legal activities, and two activities where agents took illegal maneuvers).

The main parameter that needs to be set prior to experiments is the size of the grid cells. Theoretically, smaller grid cells produce a better result with the cost of requiring more data. Based on the performed experiments, the cell size was set for the CROSS to $40 \times 25$ and for the Lankershim into $10 \times 10$ pixels. These choices of cell size divide the CROSS and Lankershim into 9 by 19 and 16 by 20 equal sized grid cells respectively. Each raw trajectory was converted into bag-of-grid representation mentioned in the section of Trajectory Analysis with Distance Dependent CRP. The dimensions of bag-of-grids representations are $X_i \in \Re^{1 \times 171}$ and $X_i \in \Re^{1 \times 320}$ for CROSS and Lankershim datasets respectively.

The correct clustering rate (CCR) is used to evaluate the clustering performance. The CCR has been used as evaluation criteria to verify trajectory clustering algorithms in several studies (*Morris & Trivedi, 2009*; *Weiming et al., 2013*; *Zhang, Kaiqi & Tieniu, 2006*). Given the ground truth set G and resulting clusters set C, corresponding cluster that maximizes

A

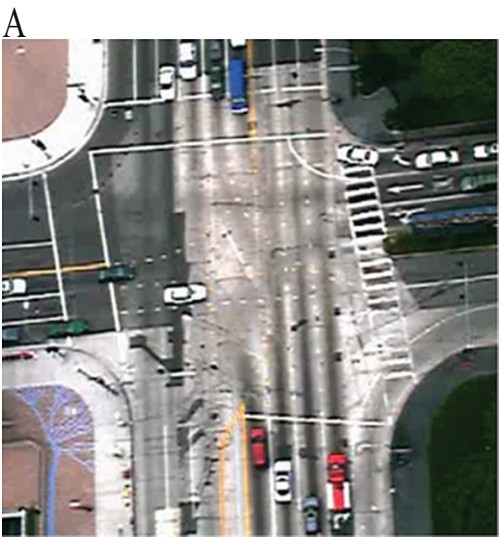

B

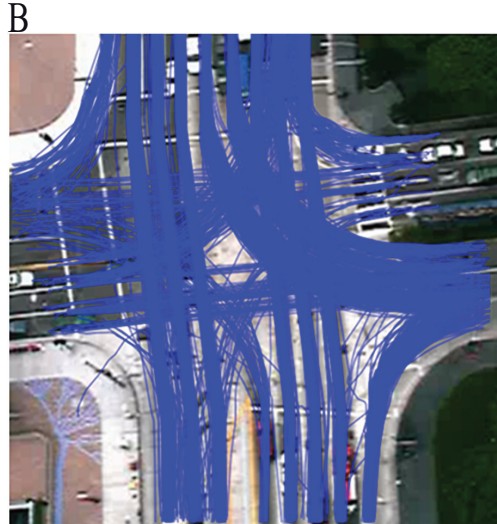

**Figure 2** **The Lankershim dataset: (A) area of interest, (B) vehicle trajectories in the interest area collected from 8:30 Am to 8:45 Am.**

the number of matched labels is found. The CCR is defined as

$$CCR = \frac{1}{N} \sum_{i=1}^{K} p_i \tag{7}$$

where $N$ is the number of trajectories, $K$ is the number of clusters in the ground truth. Given the assignment between ground truth and estimated cluster labels, $p_i$ is computed as (*Zhang, Kaiqi & Tieniu, 2006*):

$$p_i = \begin{cases} |c_i \cap g_m|; & \text{given } c_i \in C \text{ assigned to } g_m \in G \\ 0; & \text{otherwise} \end{cases} \tag{8}$$

The proposed method was compared with dual-HDP and three well-known distance measure methods, LCSS, DTW, and modified Hausdorff (MH). For each distance, four unsupervised clustering algorithms were used: K-mean clustering, spectral clustering, agglomerative clustering, and graph-based clustering. The average CCR of clustering algorithms for each distance method is reported in this study. One limitation of distance-based clustering techniques is that they require the number of clusters to be given to them.

To show the effect of choosing the number of clustering on the performance the experiments were run with the different number of clusters, including the true value. The other parameters of competitor methods were set during the course of experiments to achieve their maximum accuracy. For the proposed methods, collapsed Gibbs was performed for 100 samples. After each sampling, CCR was evaluated based on the customer assignment result. Figure 3 shows CCR per sample for the Lankershim and CROSS datasets.

A

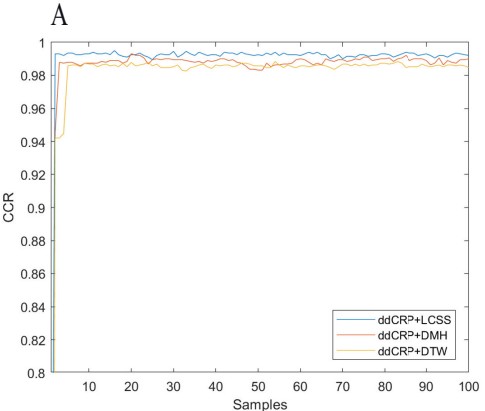

B

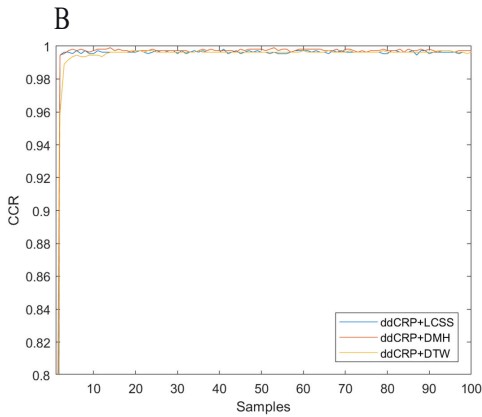

**Figure 3** Clustering accuracy of (A) the CROSS dataset, (B) the Lankershim dataset.

**Table 1** The CCR Performance of different methods for the CROSS Dataset.

| Number of Clusters | 5 | 10 | 15 | 19 | 20 | 21 | 25 | 30 |
|---|---|---|---|---|---|---|---|---|
| DTW | 0.292 | 0.559 | 0.806 | 0.971 | 0.984 | 0.968 | 0.916 | 0.857 |
| LCSS | 0.291 | 0.555 | 0.805 | 0.986 | 0.971 | 0.952 | 0.864 | 0.792 |
| MH | 0.556 | 0.559 | 0.807 | 0.986 | 0.986 | 0.973 | 0.934 | 0.879 |
| Dual HDP | – | – | – | – | – | 0.801 | – | – |
| DDCRP (DTW) | – | – | – | 0.986 | – | – | – | – |
| DDCRP (LCSS) | – | – | – | 0.993 | – | – | – | – |
| DDCRP (MH) | – | – | – | 0.989 | – | – | – | – |

In all methods, CCR achieves greater than 0.9 after the 3rd sample. The average CCR is obtained by averaging the CCR values after neglecting the first ten samples.

The results of trajectory clustering accuracy for the CROSS dataset are summarized in Table 1. The best correct clustering rate is obtained by DDCRP when using LCSS as a distance measure which produces 0.993. The average correct clustering rate of LCSS with traditional clustering algorithm is 0.986. While this value is slightly less than the performance produced by LCSS and DDCRP, it needs to be highlighted that traditional clustering techniques achieved 0.986 correct clustering rate with the assumption of knowing the true total number of clusters. Also, the proposed method improves the correct clustering rate regardless of which similarity method is used. In other words, using DTW and MH as similarity measure along with DDCRP achieve better average CCR compared to traditional clustering algorithms.

Similarly, Table 2 summarizes the clustering accuracy for the Lankershim dataset. Using DDCRP along with MH distance produces the best correct clustering rate of 0.998. Same as CROSS dataset, the proposed method improves correct clustering rate regardless of which similarity measure is used. The most notable improvement is when DTW is used as

**Table 2** The CCR Performance of different methods for the Lankershim dataset.

| Number of Clusters | 5 | 10 | 15 | 18 | 19 | 20 | 25 | 30 |
|---|---|---|---|---|---|---|---|---|
| DTW | 0.453 | 0.705 | 0.901 | 0.864 | 0.868 | 0.864 | 0.828 | 0.789 |
| LCSS | 0.529 | 0.846 | 0.901 | 0.924 | 0.925 | 0.931 | 0.912 | 0.899 |
| MH | 0.488 | 0.840 | 0.973 | 0.985 | 0.977 | 0.974 | 0.937 | 0.902 |
| Dual HDP | – | – | – | 0.974 | – | – | – | – |
| DDCRP (DTW) | – | – | – | – | 0.996 | – | – | – |
| DDCRP (LCSS) | – | – | – | – | 0.996 | – | – | – |
| DDCRP (MH) | – | – | – | – | 0.998 | – | – | – |

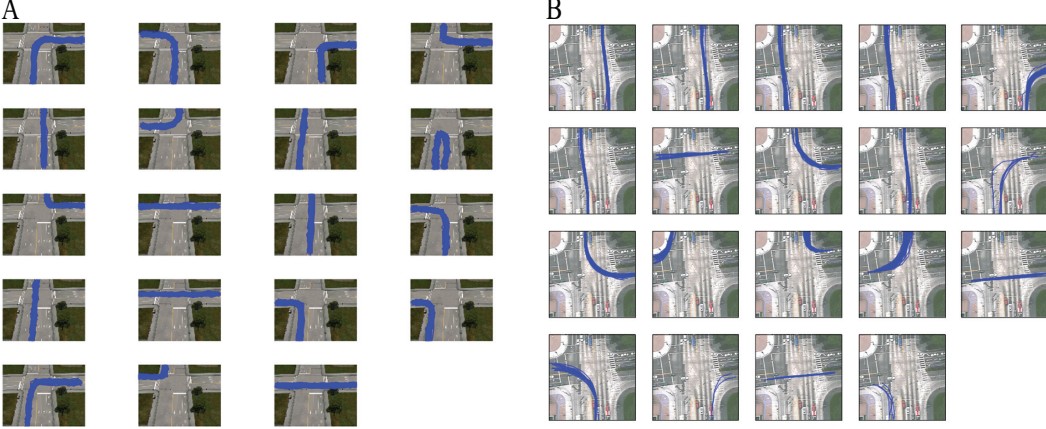

**Figure 4** Founded clusters: (A) the CROSS dataset by using the DDCRP and the LCSS distance methods, (B) the Lankershim dataset by using the DDCRP and the MH distance methods.

a similarity measure. In this case, the average CCR for similarity-based clustering is 0.868 while the combination of DTW and DDCRP results in the CCR of 0.996.

After removing clusters with single trajectory and ignoring the initial samples, methods based on DDCRP discovered 19 clusters for both the CROSS and the Lankershim datasets. Figure 4 shows the discovered clusters in the 100th sample for the CROSS and Lankershim datasets. The results shown in this figure are obtained by DDCRP using LCSS and MH distances for the CROSS and Lankershim respectively. The discovered clusters are typical activities in an intersection and include crossing the intersection, turning left, turning right, and u-turn.

As discussed in the Trajectory Analysis with Distance Dependent CRP section, the size of the grid impacts the accuracy of any PTM-based trajectory analysis system. Another advantage of the proposed method compared to the Dual-HDP method is that it is less sensitive to the choice of grid size. This is due to the fact that most PTM models, including dual HDP, are based only on bag-of-grids representation of the trajectories. The proposed method, however, uses both bag-of-grids and pairwise distance between raw trajectories. Therefore, it can be expected that the proposed method is less sensitive to the choice of grid sizes.

A

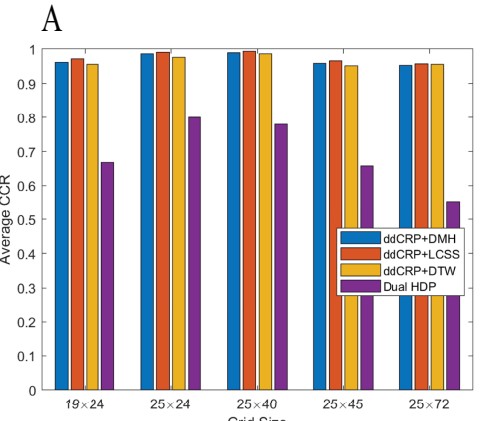

B

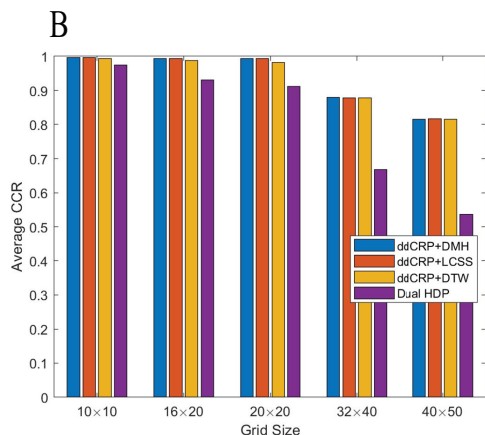

**Figure 5** The impact of grid size on the clustering accuracy: (A) the CROSS dataset, (B) the Lankershim dataset.

Figure 5 shows the average the CCR of the DDCRP and dual-HDP systems for different sizes of the grid. The grid size of $25 \times 24$ and $10 \times 10$ pixels produces 0.801 and 0.974 correct clustering rate for the dual-HDP method in the CROSS and Lankershim datasets respectively. However, the accuracy substantially decreases by increasing or decreasing the grid size. The proposed method, however, is more robust to the choice of grid size since the pairwise distance between trajectories is independent of the choice of grid size.

The aim of trajectory path modelling is to discover the paths commonly taken by objects in each cluster. One benefit of our method is its ability to model the path simultaneous to trajectory clustering. In our study, each path is characterized by the distribution over grid cells in a scene. Each cell for a path can be associated to any number in the range of 0 to 1, where 0 are the cells that have no chance of being observed in that path. As the values of a cell are closer to 1, this cell become more essential for the path, and the probability of it being passed by trajectories belonging to that path increases.

The path modelling experiments were conducted with the same parameter setup discussed earlier in this section. Figure 6 shows the cluster models for the CROSS and Lankershim datasets. The blue cells are less likely to be observed by trajectories in that cluster. Conversely, the red cells are more probably observed by trajectories. Then most paths have their probable grid cells in the middle of their route, while when moving further away to the edges of the routes, the probability of grid cells decreases.

## CONCLUSION

This paper proposed an unsupervised approach for trajectory clustering and modelling. The generative process of trajectory analysis was modelled via a probabilistic model. The pairwise distances were used as prior in DDCRP to promoting similar trajectories to be clustered. The DDCRP were used to combine the advantages of similarity-based and PTM-based approaches. Compared to probabilistic topic approaches, our method is able to model the full path taken by agents in each cluster. Unlike most similarity-based methods,

A          B

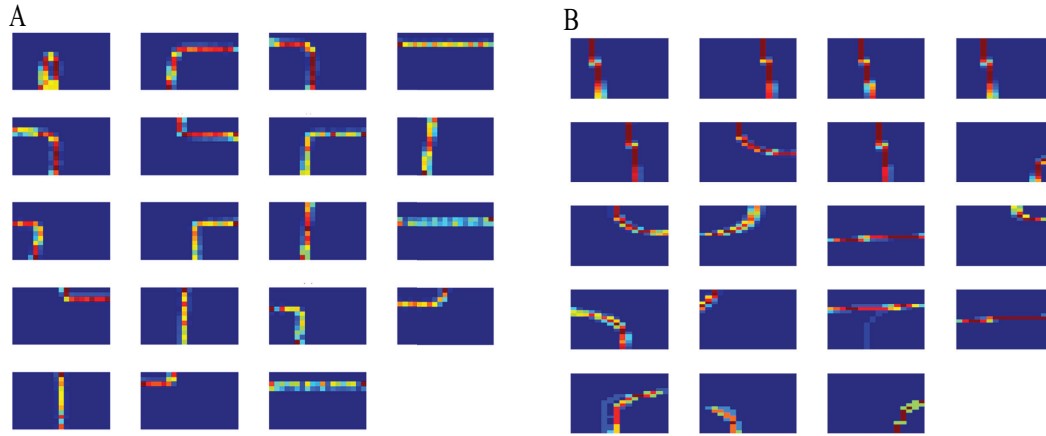

**Figure 6**   Cluster models: (A) the CROSS dataset, (B) the Lankershim dataset.

our method drives the number of clusters automatically. The proposed trajectory analysis system clusters the trajectories and models the clusters' paths at the same time. Specifically, raw trajectories were converted to bag-of-grid cells representation and considered each cluster with its distribution over the grids. Experimental results confirmed the effectiveness and usefulness of the proposed algorithm in trajectory clustering and modelling compared to other methods. The proposed approach is planned to have an online learning capability, where the cluster and path models keep updated as more data is observed.

### Funding

This work was supported by the Ministry of Education Malaysia by through a Research University Grant of University Technology Malaysia (UTM), project titled "Intelligent fault detection and diagnosing for process plant R.k430000.77434.4J010." The funders had no role in study design, data collection and analysis, decision to publish, or preparation of the manuscript.

### Grant Disclosures

The following grant information was disclosed by the authors:
Ministry of Education Malaysia by through a Research University Grant of University Technology Malaysia (UTM).

### Competing Interests

The authors declare there are no competing interests.

### Author Contributions

- Reza Arfa, Rubiyah Yusof and Parvaneh Shabanzadeh conceived and designed the experiments, performed the experiments, analyzed the data, contributed reagents/materials/analysis tools, prepared figures and/or tables, performed the

computation work, authored or reviewed drafts of the paper, approved the final draft, the authors contributed equally to the completion of the manuscript.

## Data Availability

The code is available at: https://github.com/rezaarfa/motionlearning and in the Supplemental Information.

The CROSS dataset can be downloaded from http://cvrr.ucsd.edu/bmorris/datasets/dataset_trajectory_analysis.html. The Lankershim is part of the NGSIM dataset and can be downloaded from https://ops.fhwa.dot.gov/trafficanalysistools/ngsim.htm.

## Supplemental Information

Supplemental information for this article can be found online at http://dx.doi.org/10.7717/peerj-cs.206#supplemental-information.

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
