# Peer review of "Novel trajectory clustering method based on distance dependent Chinese restaurant process"

_PeerJ Computer Science, doi:10.7717/peerj-cs.206_

## Round 0.1 · original submission · Major Revisions

Although the manuscript presents an interesting application of Chinese restaurant processes to tracking, the reviewers found major issues. In particular, the writing needs to be thoroughly revised, missing references should be included and discussed in detail, and additional experiments should be conducted, for this material to be suitable for publication.

Reviewer 1 ·

Basic reporting

The text must be thoroughly revisited as there are several and recurrent typos, specially regarding prepositions, verb conjugation and false friends; while the overall ideas can be understood, the text should definitely be improved before publication. Authors should take special care with the paragraphs between lines 185-192 and 230-233 as I can't really understand most of the ideas presented in those lines.

Typos and other grammar errors can be found in lines:
29,31,38,43,48,50,101,140,143,149,152,195-196,200,207,208,219,223,226,249,271,272,276,305,325 and figure 5 caption.

The are no major flaws on the figures and tables but the values on the y axis for figures 3 and 4 could be confusing as they go beyond 1.0, please relocate the legend.

While the authors make an a effort to either provide a straight derivation or references for most of the relevant equations in the paper, equations 8 and 9 cannot be trivially derived, and the authors provide no references for further reading.

The introduccion does a good job on highlighting the improvements of the proposed method vs other classic methods and its actual scope, but could be further improved by briefly mentioning that the paper addresses the trajectory analysis problem for traffic intersections. The abstract can also be further enhanced with this information and some of the most relevant quantitative results.

Experimental design

Overall, I have three main concerns regarding the experimental setup, and the empirical validations.

First, I would like to know why the authors choose 3 distance metrics (DTW,LCSS and MH), and 3 clustering algorithms (k-means,Spectral clustering and agglomerative clustering) after they present several more in the introduction, was the selection based on the performance of the methods?, its similarity?, another reason?. In other words, I would like to know better why the selected methods are an appropriate baseline given the author's proposal and the available data.

Second, I think the paper lacks on empirical validation to further assess the effectiveness of the proposed method, first of all I would like know why the authors use a fixed number of clusters (fixed at 19), how does the proposed method perform if that number varies?, is it optimal?, how do the baselines perform if that parameter change?, also I would like to know more about the indicator function of equation 3, and its related threshold, these two parameter seem to be critical at selecting the number of clusters but are not reported for the experimental setup could the authors perform a sensitivity analysis on the threshold or perhaps try several functions? To complement the empirical validation of the methods, could the authors evaluate the capacity of their strategy to predict the actual amount of clusters in a dataset? is it possible to use the proposed strategy to classify the activities on the Lankershim datased, and compare against the baseline?

Third, in the introduction the authors discuss about ‘probabilistic reasoning’, where a soft-assignment could be obtained as probability distribution every sample and every cluster, this possible advantage of the proposed method is never explored or validated in the paper.

Finally, as stated in amigo at all 2009, the purity metric penalizes the noise in clusters, an can yield very high values for trivial assignations, the authors should complement their empirical evaluation with metrics like Inverse purity or Van Rijsbergen F-Measure for a better understanding of the proposed method effectiveness

Validity of the findings

No comments

Reviewer 2 ·

Basic reporting

This paper presents a direct application of the distance dependent Chinese restaurant process to a tracking problem. There is no technical contribution, which is ok as long as the proposed application is novel and experimental results are convincing.

Yet, the article as it is has very poor quality in terms of writing (poorly written, unclear sentences, missing references), and experimental results (1 single dataset, insufficient comparisons with other state-of-the art methods). The article as it is needs considerable more work.

1) English: the text is very poorly written (all except the description of the DDCRP prior), I have annotated too many English typos/grammatical errors, which makes the article hard to read. Very often, there is no verb in the sentence.
Some examples:
obtained by keep tracking --> by keeping track
probabilistic reasoning can use to quantify --> can be used to quantify
which the knowledge of one helps --> ?
I suggest the authors to seek proof-reading from native English speakers.

2) The authors have missed important references, e.g., works that use particle filtering (one of the state-of-the-art techniques for trajectory modeling), as well as other Bayesian nonparametric models for tracking, such as:
A Bayesian Nonparametric Approach to Modeling Mobility Patterns (Joseph et.al, 2010)
Hierarchical Dirichlet processes for tracking maneuvering targets (Fox et.al, 2007)
Please include these references and compare your methods with some of these alternatives.

Minor:

- please number sections
- I have not found the raw data in the supplementary file. Also, demo.m does not run. Please document the raw data and code appropriately.

Experimental design

The authors use a multinomial likelihood to model continuous data, which requires quantization of the images using grids.

- How does the DDCRP compares against a (Joseph et.al, 2010)? This would be a very interesting comparison.

- How many MCMC chains were used? How was convergence assessed? Is 100 samples enough? Do the authors perform averaging over multiple samples?
The authors should run the models using different initializations and multiple chains.

Validity of the findings

- Results are inconclusive as they are. Since the main application is a direct application of the ddCRP to tracking, the authors should provide simulations with more than a single dataset. Also, the authors should compare with other probabilistic models like (Joseph et.al, 2010) to be fair and conclusive.

- Data is not provided in the supplementary (the folder /Software/demo/Lankershim/data/ is empty; there are some .mat files in Lankershim/ddcrp, but without any documentation.

Details about the dataset should be listed (how many images per label, number of pixels, etc...

---

## Round 0.2 · Minor Revisions

The revised manuscript represents a substantial improvement with respect to the original submission and addresses most of the reviewers' comments.

Please address the minor revisions from Reviewer 1.

Reviewer 1 ·

Basic reporting

There are a few typos on the manuscript.
The authors provide a fair amount of related work along clear and concise explanations of the objectives, contributions and core methods used in the paper.
Some equations (7,8,9) are better suited for an appendix, while they present a detailed analysis of an step in the methodology they not part of the author contributions.

Experimental design

The authors evaluate in two datasets for trajectory analysis in intersections crosses. Key hyperparameters of the problem as sample size, grid size and cluster amount, are thoroughly evaluated. Several baseline methods are benchmarked and compared against the proposed methodology.
Figures provide interesting insights on the trajectories discovered by the method.
Method explanations (although lengthy) provide valuable information for the replication of the proposed method.

Validity of the findings

My biggest concern on this paper is the result presentation, while the proposed method consistently outperforms the baselines across operations regimes, sometimes the improvement looks minimal (e.g. 0.986 to 0.993), I think the authors could further highlight why it is important to show improvements in a problem whose baseline is so close to perfection.
I also would like to know why the abnormal trajectories are discarded.
Other than this the conclusions are, in my opinion, validated by the presented results.

Additional comments

I think the article presents a clear methodology to improve the baseline on the problem of trajectory discovery and clustering, it is interesting that the proposed DDCRP automatically finds a number of clusters that produce improved results.
There are some minor flaws that must be revisited, but I'm leaning towards acceptance of the manuscript.

---

## Round 0.3 · accepted · Accept

The manuscript is ready for publication